# Molecular Genetics and Cytotoxic Responses to Titanium Diboride and Zinc Borate Nanoparticles on Cultured Human Primary Alveolar Epithelial Cells

**DOI:** 10.3390/ma15072359

**Published:** 2022-03-22

**Authors:** Hasan Türkez, Mehmet Enes Arslan, Arzu Tatar, Özlem Özdemir, Erdal Sönmez, Kenan Çadirci, Ahmet Hacimüftüoğlu, Bahattin Ceylan, Metin Açikyildiz, Cigdem Yuce Kahraman, Fatime Geyikoğlu, Abdulgani Tatar, Adil Mardinoglu

**Affiliations:** 1Department of Medical Biology, Faculty of Medicine, Atatürk University, 25240 Erzurum, Turkey; hturkez@atauni.edu.tr (H.T.); ahmetbceylan@icloud.com (B.C.); 2Department of Molecular Biology and Genetics, Faculty of Science, Erzurum Technical University, 25240 Erzurum, Turkey; enes.aslan@erzurum.edu.tr (M.E.A.); ozlem.ozdemir@erzurum.edu.tr (Ö.Ö.); 3Department of Otorhinolaryngology, Faculty of Medicine, Atatürk University, 25240 Erzurum, Turkey; arzutatar@atauni.edu.tr; 4Advanced Materials Research Laboratory, Department of Nanoscience & Nanoengineering, Graduate School of Natural and Applied Sciences, Atatürk University, 25240 Erzurum, Turkey; esonmez@atauni.edu.tr; 5Department of Internal Medicine, Erzurum Regional Training and Research Hospital, Health Sciences University, 25240 Erzurum, Turkey; doktorcadirci@hotmail.com; 6Department of Medical Pharmacology, Medical Faculty, Atatürk University, 25240 Erzurum, Turkey; ahmeth@atauni.edu.tr; 7Department of Chemistry, Faculty of Science and Art, Kilis 7 Aralık University, 79000 Kilis, Turkey; macikyildiz@kilis.edu.tr; 8Department of Medical Genetics, Medical Faculty, Atatürk University, 25240 Erzurum, Turkey; cigdem.kahraman@atauni.edu.tr (C.Y.K.); abdulgani@atauni.edu.tr (A.T.); 9Department of Biology, Faculty of Arts and Sciences, Atatürk University, 25240 Erzurum, Turkey; fgeyik@atauni.edu.tr; 10Science for Life Laboratory, KTH-Royal Institute of Technology, SE-17121 Stockholm, Sweden; 11Centre for Host-Microbiome Interactions, Faculty of Dentistry, Oral & Craniofacial Sciences, King’s College London, London SE1 9RT, UK

**Keywords:** titanium diboride, zinc borate, nanoparticles, in vitro, human primary alveolar epithelial cell cultures, toxicogenomic responses

## Abstract

Titanium diboride (TiB_2_) and zinc borate (Zn_3_BO_6_) have been utilized in wide spectrum industrial areas because of their favorable properties such as a high melting point, good wear resistance, high hardness and thermal conductivity. On the other hand, the biomedical potentials of TiB_2_ and Zn_3_BO_6_ are still unknown because there is no comprehensive analysis that uncovers their biocompatibility features. Thus, the toxicogenomic properties of TiB_2_ and Zn_3_BO_6_ nanoparticles (NPs) were investigated on human primary alveolar epithelial cell cultures (HPAEpiC) by using different cell viability assays and microarray analyses. Protein-Protein Interaction Networks Functional Enrichment Analysis (STRING) was used to associate differentially expressed gene probes. According to the results, up to 10 mg/L concentration of TiB_2_ and Zn_3_BO_6_ NPs application did not stimulate a cytotoxic effect on the HPAEpiC cell cultures. Microarray analysis revealed that TiB_2_ NPs exposure enhances cellular adhesion molecules, proteases and carrier protein expression. Furthermore, Zn_3_BO_6_ NPs caused differential gene expressions in the cell cycle, cell division and extracellular matrix regulators. Finally, STRING analyses put forth that inflammation, cell regeneration and tissue repair-related gene interactions were affected by TiB_2_ NPs application. Zn_3_BO_6_ NPs exposure significantly altered inflammation, lipid metabolism and infection response activator-related gene interactions. These investigations illustrated that TiB_2_ and Zn_3_BO_6_ NPs exposure may affect different aspects of cellular machineries such as immunogenic responses, tissue regeneration and cell survival. Thus, these types of cellular mechanisms should be taken into account before the use of the related NPs in further biomedical applications.

## 1. Introduction

Nanotechnology-based products have improved performance with an extended shelf life, but they come with public health concerns because of uncertain risks parallel to the development of the nanotechnological structures. Industrial areas accept that nanotechnology can improve the utilization areas of molecules in a variety of ways. On the other hand, society has many concerns about nanomaterial application due to the lack of understanding in different fields related to nanoparticle uses such as nuclear medicine and genetic modifications [1,2,3]. It is vital to increase public perceptions of risks and benefits to enable people to comprehend them in order for the field of nanotechnology to increase the social acceptance of emerging technologies [4]. Besides, there are many different molecules that can be used in nanotechnology fields such as ceramic nanoparticles, semiconductors, metal nanoparticles and polymeric nanoparticles. Each day, novel nanoparticles have been discovered and safety analyses for these compounds have been performed to enable their use in specific industries, from food sectors to medical applications [5].

Titanium diboride (TiB_2_) is known as ultrahigh-temperature ceramics (UHTCs) and belongs to the refractory transition metals family [6]. TiB_2_ exerts many favorable features such as excellent thermal conductivity, a high melting point, high hardness, high strength and good wear resistance [7,8,9]. Because of these properties, TiB_2_-based materials receive much attention from many industrial fields such as armor, cutting tools, high temperature structural materials, wear parts and electrode production [10]. TiB_2_ compounds can be found in corrosion-resistant electrical switch interfaces, ferrous metal polishing tools, aluminum extraction and scratch-resistant coating applications that are desired features for industrial and technological machineries [11]. Such compounds are often desired to have non-toxic properties to allow their full potential in a variety of applications and avoid human health risks. The biocompatibility of titanium–titanium boride (Ti-TiBw) composites for use in medical and dental implants was investigated with the results indicating that Ti-TiBw composite is blood biocompatible and not cytotoxic to fibroblast cells [12].

Zinc (Zn) molecules are essential components for cellular mechanisms such as the antioxidant response, the glutaminergic transmission in the brain, transcriptional regulations and enzyme activity. Zn deficiency can emerge with the aging process and the absence or reduction of the Zn element generally results in brain abnormalities, sensory dysfunctions, immunodeficiency and growth retardation [13,14,15]. Furthermore, Zinc borate (Zn_3_BO_6_) can used in different areas, for instance, smoke suppressants, antitracking agents and flame extinguishers [16]. Nano-sized Zn_3_BO_6_ molecules have been shown to possess less negative effects and higher flame retardancy compared to the ordinary flame extinguisher with homogenic dispersions and higher flame resistance [17]. In addition, an in vitro cytotoxicity analysis of zinc fructoborate (ZnFB) executed a low cytotoxic property and the use of this boron-containing compound was suggested as a nutritional molecule for Zn^2+^ supplementation instead of zinc orotate [18]. Moreover, in regards to providing modest antimutagenic potential, zinc borate was discovered to have moderate total antioxidant activity. With its antioxidative and antimutagenic qualities, zinc borate has the potential to be used in medicine [19].

Despite substantial research on these compounds, research into TiB_2_ and Zn_3_BO_6_ compounds for in vitro biological activities is limited and, to the authors’ knowledge, nothing is known about their biological properties and cytotoxic activities, which has prevented their usage in bio-applications thus far. In the current study, TiB_2_ or Zn_3_BO_6_ NPs were characterized by using X-ray crystallography (XRD), transmission electron microscope (TEM), scanning electron microscope (SEM) and energy-dispersive X-ray spectroscopy (EDX) analyses and also tested for in vitro cytotoxic damage potentials on human primary alveolar epithelial cell cultures (HPAEpiCs) by different viability assays (MTT, LDH release and NR uptake cell viability assays). In addition, a whole-genome microarray analysis was performed to determine the differentially expressed genes in TiB_2_- or Zn_3_BO_6_ NPs-treated HPAEpiC cells. 

## 2. Materials and Methods

### 2.1. Characterizations of Nanoparticles

Titanium diboride (TiB_2_) nanoparticles (Sigma-Aldrich^®^, St. Louis, MO, USA) were obtained commercially and particle characterization and verifications were performed. Zinc borate (Zn_3_BO_6_) nanoparticles were synthesized in a batch reactor by mixing 25 mL of distilled water, 7.5 g of boric acid (H_3_BO_3_, Sigma-Aldrich^®^, St. Louis, MO, USA), 2.846 g of zinc oxide (ZnO, Sigma-Aldrich^®^, St. Louis, MO, USA), 10 mL of pure ethanol (Sigma-Aldrich^®^, St. Louis, MO, USA) and 3g of oleic acid (Sigma-Aldrich^®^, St. Louis, MO, USA) in a 250 mL three-necked round-bottom container. Reaction was initiated by heating the mixture at 95 °C for 4 h. After the reaction period ended, white precipitates were filtrated and transferred to a fresh container, and the sample was washed 6 times with pure ethanol and water, sequentially. Samples were dried at 80 °C and nano-sized Zn_3_BO_6_ were characterized by using SEM, TEM and XRD techniques. 

The structural analysis of TiB_2_ and Zn_3_BO_6_ nanoparticles were operated by using X-ray diffraction (XRD) via the use of a Rigaku diffractometer (Houston, TX, USA) with CuKα radiation operated at 40 kV and 30 mA at room temperature. Observations were performed with step of 0.020 in the geometry of coupled θ-2θ changed between 40 and 400, respectively. Particle sizes and surface morphologies of nanoparticles were investigated under a scanning electron microscope (FEI inspect S50 SEM, Hilsboro, OR, USA) and transmission electron microscopy (JEOL JEM-ARM200CFEG UHR-TEM, Tokyo, Japan).

### 2.2. Cell Cultures and NPs Applications

The human primary alveolar epithelial cell cultures (HPAEpiC, ScienceCell^®^, Carlsbad, CA, USA) were grown in 500 mL of alveolar epithelial cell medium (AEpiCM, ScieneCell^®^, Carlsbad, CA, USA), including 5 mL of penicillin/streptomycin solution (Sigma-Aldrich^®^, St. Louis, MO, USA), 5 mL of epithelial cell growth supplement and 10 mL of fetal bovine serum (Sigma-Aldrich^®,^ St. Louis, MO, USA). Cell cultures were grown in a humidified incubator (ThermoFisher^®^, Waltham, MA, USA) with 5% CO_2_ at 37 °C until cell densities reached 80% confluency. A total of 10 mg of each nanoparticle was placed into polypropylene tubes and dispersed in Milli-Q water (10 mL, at 25 °C, Merck, Darmstadt, Germany) using a sonification chamber (Spectra^®^, Carson, CA, USA). The total sonication time was 40 min (ultrasonication was performed for 20 min). To prevent heating during ultrasonic dispersion the tubes were kept at 0 °C in an ice-water bath. After autoclaving of 10 mg/mL NP/Milli-Q dispersions, the stock dispersions were stored in the dark at autoclaved and stored as stock dispersions at 4 °C. TiB_2_ and Zn_3_BO_6_ NPs were applied to the cell cultures for 72 h at a wide spectrum of concentrations (0.625 mg/L to 1280 mg/L) in triple repeats. Hydrogen peroxide (H_2_O_2_; 25 μM, Sigma-Aldrich^®^, St. Louis, MO, USA) was added to cultures as the positive control for cytotoxicity analyses. 

### 2.3. Cytotoxicity Testing

#### 2.3.1. 3-(4,5-Dimethylthiazol-2-yl)-2,5-diphenyltetrazolium Bromide (MTT) Assay

For cell viability analyses 3-(4,5-Dimethylthiazol-2-yl)-2,5-diphenyltetrazolium bromide (MTT, ThermoFisher^®^, Waltham, MA, USA) assay was performed in NPs applications. MTT solution was prepared according to the manufacturer instructions and applied to cell cultures in 5 µM concentrations. Cell cultures were incubated at 37 °C and 5% CO_2_ for 3 h to produce formazan crystals. Dimethyl sulfoxide (DMSO, Sigma-Aldrich^®^, St. Louis, MO, USA) was used to solve formazan crystals to obtain a blue-purple color as a cell viability indicator. A microplate reader (Bio-Tek Instruments, Winooski, VT, USA) was used to measure color intensities at 570 nm wavelength. 

#### 2.3.2. Lactate Dehydrogenase (LDH) Release Assay

Cytotoxicity in cell cultures was investigated by using lactate dehydrogenase (LDH) assay. LDH assay (Cayman Chemical Company^®^, Ann Arbor, MI, USA) was performed in accordance with the manufacturer’s instructions. Different concentrations of NPs were applied to cell cultures for 72 h in 48-well plates. A total of 100 µL of cell culture supernatants were transferred to fresh cell culture plates and 100 µL of reaction mixture was added to the cultures, and incubated for 30 min at room temperature in a dark place. Color intensities were measured by using a microplate reader at 490 nm absorbance.

#### 2.3.3. Neutral Red (NR) Uptake Assay

Neutral red (NR, Sigma-Aldrich^®^, St. Louis, MO, USA) solution was applied to the cell cultures to observe cell viabilities in NP-administered cultures for 2 h at 37 °C. CaCl_2_ (0.25%, Sigma-Aldrich^®^, St. Louis, MO, USA) and formaldehyde (0.125%, Sigma-Aldrich^®^, St. Louis, MO, USA) mixture was used to remove excess NR solution in the cultures. Finally, acetic acid (1%, Sigma-Aldrich^®^, St. Louis, MO, USA) and ethanol (50%) mixture was added to the cell cultures and incubated at room temperature for 30 min to obtain red colors in each well. Color intensities were read at 540 nm absorbance by using a microplate reader.

### 2.4. Microarray Analysis

IC_50_ concentrations of NPs (Zn_3_BO_6_: 66.50 mg/L and TiB_2_: 104.34 mg/L) were applied to the cell cultures and total RNA isolation procedure was performed at the end of a 72 h duration. RNA isolations were performed by using a PureLink™ RNA Mini Kit (Invitrogene^®^, Waltham, MA, USA) and isolated RNA quality was investigated by using a bio-analyzer (Agilent Technologies, Santa Clara, CA, USA) and UV-visible spectrophotometer (NanoDrop^®^, ThermoFisher^®^, Waltham, MA, USA). A TargetAmp-Nano Labeling kit was used to amplify isolated RNAs and an Illumina Expression BeadChip (EPICENTRE, Madison, WI, USA) was utilized to obtain biotinylated cRNAs. By using a T7 oligo (dT) primer 500 ng of total RNA was reverse-transcribed to cDNA, second-strand cDNA was synthesized, in vitro transcribed, and labeled with biotin-NTP and cRNA was quantified using the ND-1000 Spectrophotometer. Human HT-12 v4.0 Expression Beadchip (Illumina Inc., SanDiego, CA, USA) hybridization was used to label cRNAs at 58 °C for 17 h. Amersham fluorolink streptavidin-Cy3 (GE Healthcare Bio-Sciences, Piscataway, NJ, USA) bead array was used to detect array signals. Array signals were read by using the Illumina bead array reader.

### 2.5. Data Analysis 

SPSS program (IBM^®^, Endicott, NY, USA), Duncan’s test was performed for each experiment for statistical calculations at a significance level ≤ 0.05. Internal quality control checks of the raw data were used to analyze overall quality of hybridizations and chip performances. Illumina GenomeStudio v2011.1 (Gene Expression Module v1.9.0, Illumina Inc., San Diego, CA, USA) software was used to gather raw data according to the manufacturer instructions. Logarithmic transformations and normalizations of array probes were performed by using the quantile method. Fold-change values were used to show gene expressions and statistically significant differences. STRING analysis (Protein-Protein Interaction Networks Functional Enrichment Analysis) was performed for gene probes to assess differentially expressed gene relationships. 

## 3. Results 

The SEM image of TiB_2_ (Figure 1a) shows that the material has a very homogeneous granular structure. Particle sizes were monitored between 100–150 nm in TEM analysis (Figure 1b). The X-ray diffraction results showed that the most dominant peak was obtained at 2θ = 35.90. In addition, the values of 34 and 31.30, respectively, in the diffraction pattern of TiB_2_ appear as the other largest dominant peaks (Figure 1c). In addition, the crystalline structure of the material is clearly seen from the peaks in the diffraction pattern in Figure 1c. SEM analysis showed variable sizes of Zn_3_BO_6_ nanoparticles and confirms that the material has a highly heterogenic structure (Figure 2a). It was obtained from TEM images that the Zn_3_BO_6_ compound has particle sizes at the nano level ranging from 50 to 80 nm (Figure 2b). The high number of peaks in the diffraction pattern of Zn_3_BO_6_ indicates that the material grows in a polycrystalline structure (Figure 2c). 

All of the performed cell viability assays (MTT, LDH and NR) indicated a clear concentration-dependent manner cytotoxicity. Boron NPs did not show cytotoxic effects at all concentrations below 20 mg/L (0.625, 1.25, 2.5, 5 and 10 mg/L). TiB_2_ and Zn_3_BO_6_ NPs started to cause cytotoxicity at concentrations of 20 mg/L and above. IC_50_ values were calculated using probit analysis from the results obtained from the MTT assay. The toxicity potentials of the compounds were ranked according to their IC_50_ values. Zn_3_BO_6_ (IC_50_: 66.50 mg/L) was shown to be less cytotoxic than TiB_2_ (IC_50_: 104.34 mg/L) (Figure 3 and Figure 4).

As a result of microarray analysis performed in cells treated with TiB_2_, it was revealed that there were changes in the expression levels of 14 genes. It was observed that five of these genes had an increase in gene expression (Fold change ≥1.5) and there was a decrease in gene expression in nine of them. After the analysis, it was observed that the expression levels of FOS, FOSB (transcription factor) and MXRA5 (cellular adhesion molecule) genes decreased in cells after TiB_2_ application. In addition, there was an increase in the expression of genes such as MMP3 (protease) and ANXA10 (carrier protein). According to STRING (Protein-Protein Interaction Networks Functional Enrichment Analysis) analysis, DNA metabolic pathways were most affected in the cell. The immune system, defense mechanisms and many transcription factor pathways were affected by TiB_2_ treatment. Considering the number of affected genes, it was observed that there were significant changes in the expressions of genes involved in the immune system (Table 1 and Figure 5). Table 1 lists the 50 genes with the highest expression levels in lung epithelial cells treated with TiB_2_ NPs. 

In the cells treated with Zn_3_BO_6_ NPs, it was determined that there were changes in the expression levels of 875 genes. Significant increase in gene expression (Fold change ≥2) was observed in 486 of these genes and a decrease in gene expression in 389. As a result of microarray analysis, it was seen that Zn_3_BO_6_ NPs is effective upon cell cycle and cell division. Different gene pathways such as responses in chromosomal regions, kinetochore organization and the regulation of extracellular matrix molecules are affected by Zn_3_BO_6_ NPs application. Again, after treatment with Zn_3_BO_6_ NPs, cell cycle in cells, cancer pathways and cytokine-receptor molecule regulation are affected. Table 2 presents 50 genes with the highest expression levels in lung epithelial cells treated with Zn_3_BO_6_ NPs.

## 4. Discussion

In the last decade, nanotechnology has been utilized in many different fields from industrial applications to food and health applications. Recently, boron-based nano-compound integrations into various technologies have begun to be discussed in many studies because of their superior and versatile properties [20,21,22,23]. Studies have claimed that inhalation is the most common route for NPs exposure and molecules are transferred from the upper respiratory tract to bronchioles by electrostatic force of the air. Thus, the HPAEpiC cell line is one of the most suitable cell culture models to study NP toxicity that can easily be contacted through the air [24]. In the present study, the toxicogenomic properties of TiB_2_ and Zn_3_BO_6_ NPs were investigated on HPAEpiC cell cultures by using three different cytotoxicity tests (MTT, LDH and NR assays) and microarray analyses to assess their applicability potentials for different areas. Furthermore, the physical characteristics of TiB_2_ and Zn_3_BO_6_ NPs were analyzed via the use of SEM, TEM and XRD techniques to determine their sizes and structures. SEM analyses put forth that commercially obtained TiB_2_ NPs had a very homogeneous granular structure and diffraction pattern peak investigations showed crystalline construction of TiB_2_ NPs. TEM investigations showed that TiB_2_ NPs had particle sizes of around 100–150 nm. Besides, the high number of peaks in the diffraction pattern of Zn_3_BO_6_ NPs indicated that the compound was grown in a polycrystalline structure. It was observed from the TEM images that Zn_3_BO_6_ NPs had particle sizes between 50 and 80 nm. 

Moreover, cell viability analyses gave similar results stating that cytotoxic concentration for both TiB_2_ NPs and Zn_3_BO_6_ NPs enhanced at 20 mg/L for HPAEpiC cell cultures. On the other hand, when IC_50_ concentrations were calculated for each NP, TiB_2_ NPs had 104.348 mg/L and Zn_3_BO_6_ NPs had 66.506 mg/L concentration which decreased 50% of cellular populations in HPAEpiC cell cultures. IC_50_ concentrations of both TiB_2_ and Zn_3_BO_6_ NPs were applied to the cell cultures for 72 h and at the end of the incubation period microarray analyses were performed by isolating total RNAs from the cell cultures. Differentially expressed gene probes were associated by using STRING (Protein-Protein Interaction Networks Functional Enrichment Analysis) analysis to understand the regulatory properties of TiB_2_ and Zn_3_BO_6_ NPs on the HPAEpiC cell line [25]. STRING analysis showed that prostaglandin-endoperoxide synthase 2 (PTGS2), matrix metalloproteinase 3 (MMP3), intercellular adhesion molecule 1 (ICAM1) and DExD-box helicase 52 (DDX52) differentially expressed gene probes were strongly correlated with other gene groups with respect to TiB_2_ NPs application (Figure 5). Furthermore, apolipoprotein E (APOE), C-C motif chemokine ligand 2 (CCL2), chitinase 3 like 1 (CHI3L1) and matrix metalloproteinase 3 (MMP3) gene probes were found to have strong relationships with other gene probes that were differentially expressed in HPAEpiC cell cultures against Zn_3_BO_6_ NPs exposure (Figure 6). 

Prostaglandin-endoperoxide synthase (PTGS), also referred to as cyclooxygenase, acts as both dioxygenase and peroxidase in prostaglandin biosynthesis. Gene regulation was stimulated by the different factors by which the biosynthesis of prostanoids can enhance mitogenesis and inflammation [26]. Furthermore, the regenerative cell survival and wound healing properties of PTGS were investigated in different tissue types [27,28,29]. In the present study, TiB_2_ application was found to increase PTGS gene expression in HPAEpiC cell cultures indicating the enhanced regenerative status. Matrix metalloproteinase (MMP) proteins play an important role in physiological processes such as reproduction, embryonic development and tissue remodeling, and are also known to be effective in disease progressions such as metastasis and arthritis [30,31]. MMP proteins were shown to be secreted as inactive forms and activated by extracellular proteinases. These enzymes were shown to degrade laminin, fibronectin, cartilage proteoglycans and various collagens. MMP enzymes were shown in different studies to have crucial roles in tissue repair, tumor initiation and atherosclerosis progression [32,33,34]. Intercellular adhesion molecule 1 (ICAM1) is a cell surface glycoprotein typically expressed in immune and endothelial cells. It binds to CD11b/CD18 and CD11a/CD18 type integrins used as viral attachments to a receptor [35,36]. Furthermore, a previous study showed that ICAM1 overexpression in mesenchymal stem cells could be a useful tool for immune-regulator dendritic cells (DCs) and T cells in vivo and in vitro [37]. DExD-box helicase 52 (DDX52) was investigated to have important roles in different cellular processes such as cellular RNA metabolism, translation, transcription and infections [38]. Moreover, DDX52 helicase was shown to regulate innate immunity against viral infections and enhance cellular processes to prevent viral replications [39,40]. 

Apolipoprotein E (APOE) was found to encode chylomicron apoproteins that interact with a specific peripheral and liver cell receptor to play an essential role for triglyceride-rich lipoprotein catabolism [41,42]. Previous studies showed that APOE mutant mice could be a good chronic obstructive pulmonary disease model exhibiting emphysema, pulmonary inflammation and airway obstruction. Furthermore, the mutant models share common risk factors with other cardiovascular diseases [43,44,45]. In our study, the APOE gene was found to be upregulated with respect to Zn_3_BO_6_ NPs application to HPAEpiC cell line. On the other hand, C-C motif chemokine ligand 2 (CCL2) was observed to be downregulated in HPAEpiC cell cultures against Zn_3_BO_6_ NPs exposure. CCL2 was investigated as a member of the chemokine superfamily involved in inflammatory and immunoregulatory processes [46,47]. It was found that the overexpression of the CCL2 gene was closely related to severe acute respiratory syndrome coronavirus 2 [48,49]. Chitinase 3 like 1 (CHI3L1) protein was shown to catalyze chitin hydrolysis and was highly expressed in fungal cell walls and insect exoskeletons [50]. Furthermore, eight human CHI3L1 gene family members were found to encode glycosyl hydrolase that was expressed in neutrophils, macrophages and chondrocytes which play important role in tissue remodeling and inflammation [51,52]. 

In the light of the observations, different cell viability assays revealed that TiB_2_ and Zn_3_BO_6_ NPs exhibited low cytotoxic properties upon HPAEpiC cell cultures for the first time. Furthermore, microarray analyses showed that differentially expressed genes with respect to TiB_2_ NPs application were closely related to inflammation, cell regeneration and tissue repair. Moreover, TiB_2_ NPs exposure was observed to upregulate gene expression associated with immunological regulations. Besides, Zn_3_BO_6_ NPs application was analyzed to have an inhibitory effect on pulmonary inflammation, regulatory properties on lipid metabolism and infection response. Both of the boron compounds were found to have common stimulatory features on HPAEpiC cell cultures such as immunological responses and tissue regeneration. To conclude, TiB_2_ and Zn_3_BO_6_ NPs were investigated as low toxic compounds when exposed to a human pulmonary alveolar cell culture and might not cause dramatic side effects under certain conditions. On the other hand, further animal studies should be performed before recommending TiB_2_ and Zn_3_BO_6_ NPs as biocompatible materials. 

## Figures and Tables

**Figure 1 materials-15-02359-f001:**
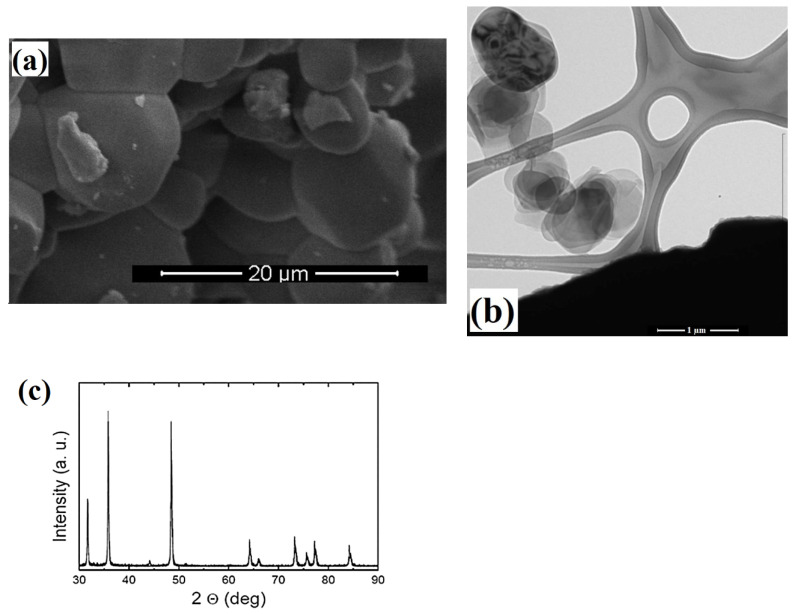
Characterization of titanium diboride (TiB_2_) NPs. (**a**) scanning electron microscope (SEM) analysis of TiB_2_ NPs, (**b**) transmission electron microscope (TEM) analysis of TiB_2_ NPs and (**c**) X-ray crystallography (XRD) analysis of TiB_2_ NPs.

**Figure 2 materials-15-02359-f002:**
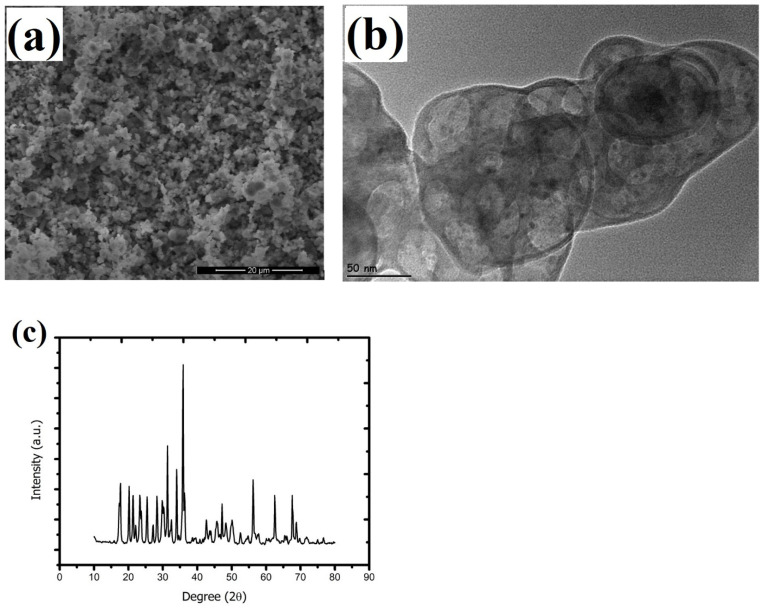
Characterization of zinc borate (Zn_3_BO_6_) NPs. (**a**) scanning electron microscope (SEM) analysis of Zn_3_BO_6_ NPs, (**b**) transmission electron microscope (TEM) analysis of Zn_3_BO_6_ NPs and (**c**) X-ray crystallography (XRD) analysis of Zn_3_BO_6_ NPs.

**Figure 3 materials-15-02359-f003:**
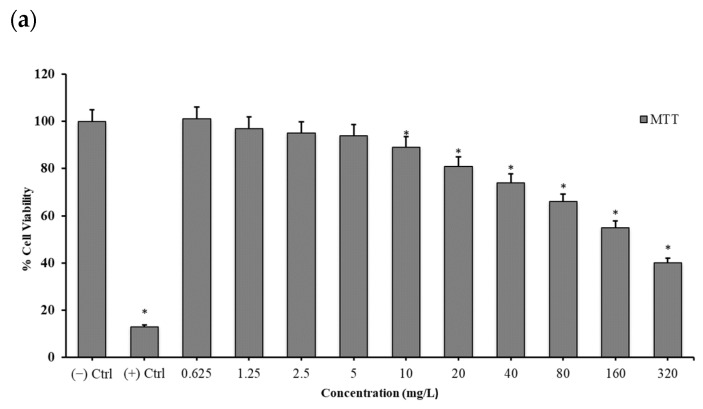
Cell viability assay of titanium diboride (TiB_2_) NPs on human pulmonary alveolar cells. (**a**) 3-(4,5-Dimethylthiazol-2-yl)-2,5-diphenyltetrazolium bromide (MTT) cell viability analysis of TiB_2_ NPs, (**b**) lactate dehydrogenase (LDH) cytotoxicity analysis TiB_2_ NPs and (**c**) neutral red (NR) cell viability analysis of TiB_2_ NPs on human pulmonary alveolar cells. * SPSS program, Duncan’s test was performed for each experiment for statistical calculations at a significance level ≤ 0.05.

**Figure 4 materials-15-02359-f004:**
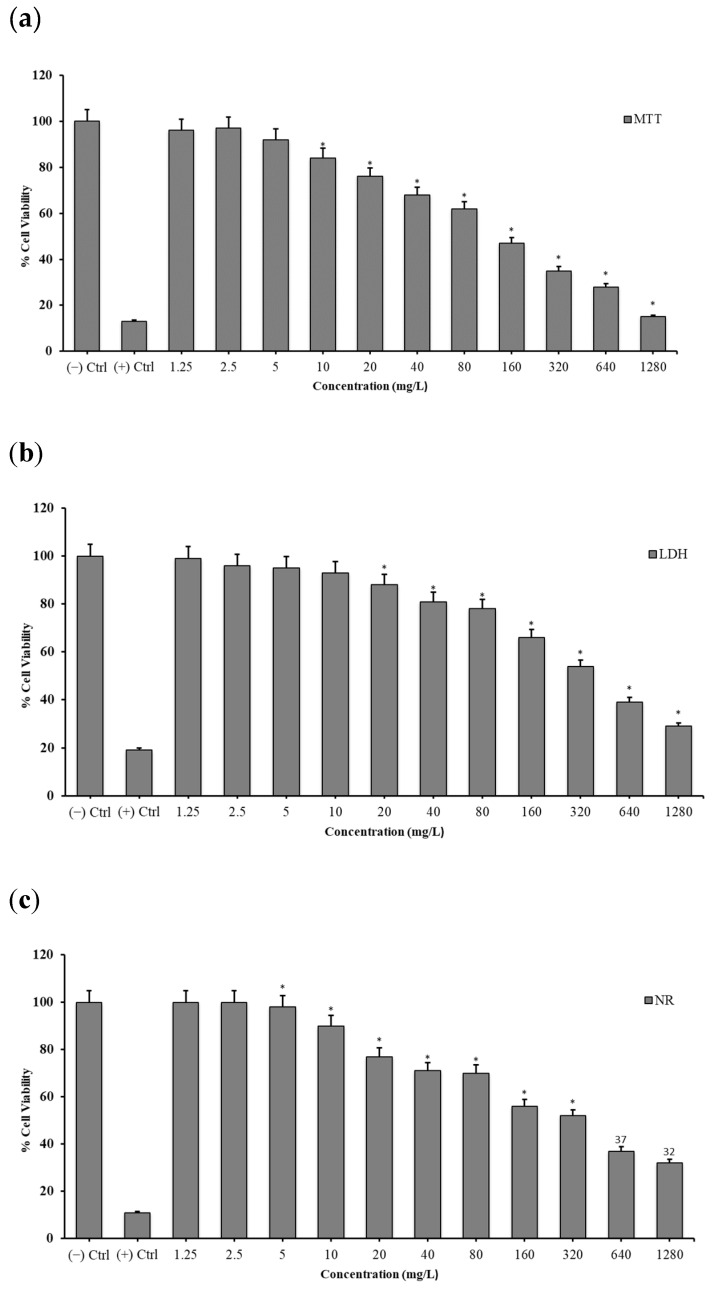
Cell viability assay of zinc borate (Zn_3_BO_6_) NPs on human pulmonary alveolar cells. (**a**) 3-(4,5-Dimethylthiazol-2-yl)-2,5-diphenyltetrazolium bromide (MTT) cell viability analysis of Zn_3_BO_6_ NPs, (**b**) lactate dehydrogenase (LDH) cytotoxicity analysis Zn_3_BO_6_NPs and (**c**) neutral red (NR) cell viability analysis of Zn_3_BO_6_ NPs on human pulmonary alveolar cells. * SPSS program (SPSS^®^, USA), Duncan’s test was performed for each experiment for statistical calculations at a significance level ≤ 0.05.

**Figure 5 materials-15-02359-f005:**
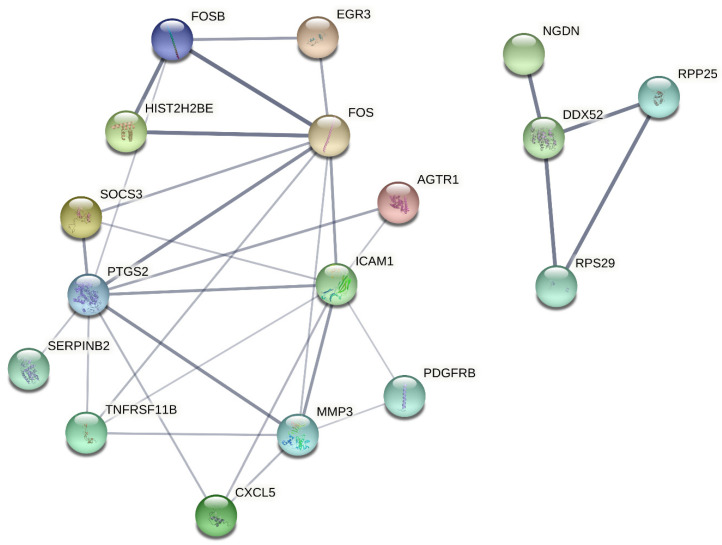
String analysis of titanium diboride (TiB_2_) NPs.

**Figure 6 materials-15-02359-f006:**
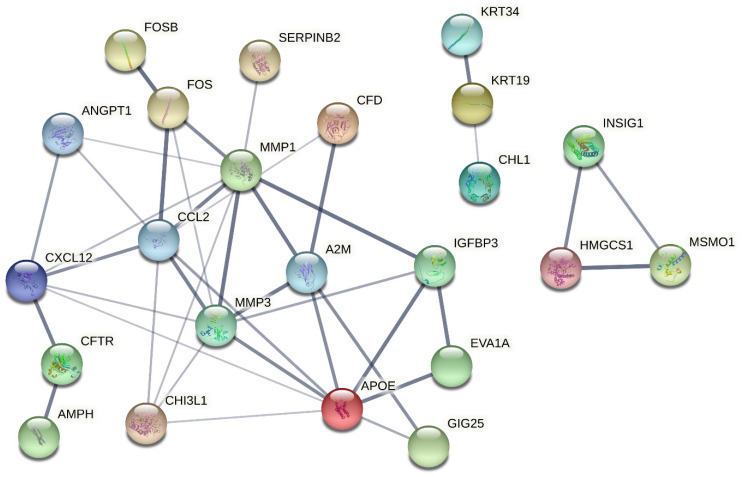
String analysis of zinc borate (Zn_3_BO_6_) NPs.

**Table 1 materials-15-02359-t001:** Gene expression changes with respect to titanium diboride (TiB_2_) NPs application.

Titanium Diboride Fold Change (FC)			
Upregulated Expressions	FC	Downregulated Expressions	FC
CXCL5	1.81	FOS	−1.88
GINS4	1.80	MGC24103	−1.69
MMP3	1.67	LOC100132564	−1.67
ANXA10	1.53	MXRA5	−1.58
PTGS2	1.50	RPS29	−1.57
PTGS2	1.47	FOSB	−1.52
BEX2	1.47	INMT	−1.51
CXCL5	1.45	LOC100134259	−1.50
NEBL	1.41	MGC4677	−1.50
SERPINB2	1.41	LOC728499	−1.47
IL13RA2	1.41	FAM65C	−1.46
DIRAS3	1.40	C7orf54	−1.45
ASNS	1.40	BAALC	−1.41
NGDN	1.39	STARD13	−1.41
CRLF1	1.38	IFIT1	−1.40
PTPRH	1.37	SOCS3	−1.39
LOC143666	1.34	PDGFRB	−1.38
ICAM1	1.34	TNFRSF11B	−1.37
AGTR1	1.33	HEYL	−1.37
LOC100008588	1.32	EGR3	−1.37
RPP25	1.32	WDR33	−1.36
DDX52	1.32	NEDD9	−1.36
LOC441253	1.32	CRIP1	−1.36
RRAD	1.32	NDRG4	−1.34
SLC8A3	1.31	HIST2H2BE	−1.34

**Table 2 materials-15-02359-t002:** Gene expression changes with respect to zinc borate (Zn_3_BO_6_) NPs application.

Zinc Borate Fold Change (FC)			
Upregulated Expressions	FC	Downregulaed Expressions	FC
APOE	22.44	NPTX1	−15.36
ADH1A	20.48	SERPINB2	−6.68
IGFBP3	16.88	ESM1	−6.55
IGFBP3	15.34	SERPINB2	−5.80
CFD	14.08	LOC728285	−5.54
PTGDS	13.67	MMP1	−4.63
MFAP4	13.41	CCL2	−4.57
RGS4	12.28	NPTX1	−4.57
SERPINA3	11.94	ESM1	−4.48
CXCL12	10.54	INSIG1	−4.45
FOS	10.25	NPTX1	−4.28
PARM1	9.60	KRT19	−4.27
AMPH	9.38	LOC100134073	−4.16
ANGPT1	8.62	KRT34	−4.12
A2M	8.47	H2AFY2	−4.09
CXCL12	8.31	LOC728255	−4.08
CHL1	7.68	GPR68	−4.04
HSD17B2	7.58	TNFSF4	−3.94
FMO2	7.50	E2F7	−3.92
CFTR	7.47	HMGCS1	−3.87
H19	7.40	RGMB	−3.77
CHI3L1	7.39	SC4MOL	−3.74
FOSB	7.37	MMP3	−3.73
CORO6	7.36	TMEM166	−3.69
FER1L4	7.32	TNFSF4	−3.68

## Data Availability

The data presented in this study are available on request from the corresponding author. The data are not publicly available due to privacy.

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
