# Peer review of "Molecular Genetics and Cytotoxic Responses to Titanium Diboride and Zinc Borate Nanoparticles on Cultured Human Primary Alveolar Epithelial Cells"

_materials, 2022, doi:10.3390/ma15072359_

Round 1
Reviewer 1 Report
In this manuscript, the authors reported preparation of TiB2 and Zn3BO6, and evaluated their toxicological properties on HPAEpiC. In particular, molecular genetics and alterations in pathological protein expression were substantially studied. In summary, this is a nice work revealing the cytotoxic molecular mechanism of TiB2 and Zn3BO6. Whereas I would recommend publication after addressing my questions below via a major revision.
- In Figure 1, color of labels for figure panels (e.g. a, b, c) should be set to black.
- For figure 1 and 2, the authors should describe clearly in figure captions what a, b, and c respectively refers to. And please don’t overlay images or plots together. Display them in a separate panel.
- TEM result of Zn3BO6NPs is of poor quality. From the current TEM and SEM results, it is hard to conclude Zn3BO6 are nanosized particles. The authors should either provide higher quality data or run other tests to validate its size.
- The authors should separate MTT/LDH/NR results in Figure 3 and figure 4, and display them in individual panels. Also, the authors should indicate clearly in the figure each P value was between which groups.
- Maybe I missed it, but the authors should describe more experimental details about microarray analysis.
Author Response
- In Figure 1, color of labels for figure panels (e.g. a, b, c) should be set to black.
Label colors were set to dark colors.
- For figure 1 and 2, the authors should describe clearly in figure captions what a, b, and c respectively refers to. And please don’t overlay images or plots together. Display them in a separate panel.
Captions were modified and images are separated into different panels.
- TEM result of Zn3BO6NPs is of poor quality. From the current TEM and SEM results, it is hard to conclude Zn3BO6 are nanosized particles. The authors should either provide higher quality data or run other tests to validate its size.
Image qualities were modified.
- The authors should separate MTT/LDH/NR results in Figure 3 and figure 4, and display them in individual panels. Also, the authors should indicate clearly in the figure each P value was between which groups.
Cell viability results were separated into different panels.
- Maybe I missed it, but the authors should describe more experimental details about microarray analysis.
Microarray analysis part was modified and more detail was added to the part.

Reviewer 2 Report
Manuscript ID: materials-1639277
This article deals with the evaluation of toxicogenomic properties of Titanium diboride (TiB2) and zinc borate (Zn3BO6) nanoparticles in human primary alveolar epithelial cell cultures (HPAEpiC). Authors performed characterization studies using various techniques such as XRD, SEM and TEM. Next, authors used 3 different cytotoxicity assays (MTT, LDH and NR uptake) to predict the toxicity profile of both NPs. Authors carried microarray analyses in HPAEpiC cells treated with TiB2 and Zn3BO6 NPs for changes in gene expression. This is an interesting study and authors performed limited but focused investigation using selective NPs. However, there are some missing aspects which must be addressed with modification in the data & text with suitable explanations.
Comments:
- This article is about ‘titanium diboride and zinc borate nanoparticles’; however there is not much information on NPs. Introduction contains information mostly about general titanium diboride and zinc borate compounds. Authors should include information on NPs than general compounds.
- Materials and Methods: In the sub-section ‘Characterization of Nanoparticles’, what is the amount of ‘pure ethanol’ and ‘oleic acid’ used?
- Materials and Methods – Cell cultures and exposure – Please mention the solvent used to dissolve the NPs.
- Materials and Methods – NR uptake assay – Please include the treatment duration of NPs to the cells.
- Materials and Methods – Microarray analyses – There is no information on the treatment duration and concentration of each NPs used for this analysis.
- XRD, SEM and TEM images in Figures 1 and 2 must be presented separately (No overlapped images). Images shown in Figures 1B and 2B are not clear. Authors must include high-resolution images for these figures. Further, authors claim ‘highly granular structure’ for the SEM images of ZN3BO6 NPs; however there is no granular structure in the image. Please correct it or replace the image.
- Authors used ‘LC50’ while presenting in vitro cytotoxicity results and the term ‘IC50’ was used in the discussion section. It should be ‘IC50’ throughout the manuscript. Also, authors should include the Figure numbers while presenting the results.
- Authors must clarify which cytotoxicity data was used to calculate LC50/IC50 values for both the NPs. It is better to include IC50 values for each NPs derived from individual cytotoxicity assays.
- ‘According to DAVID annotation….’, reference is missing. Please include suitable reference for the analysis used.
- Authors should mention Table 1 and 2 while presenting the results.
- What is the rationale for using HPAEpiC cells for this study? Please explain.
- Discussion – ‘As a conclusion, TiB2 and Zn3BO6……’, this statement needs to be modified/corrected as authors performed cytotoxicity studies in only one cell line which do not reflect complete toxicity profile of these NPs.
Minor:
- ‘…nuclear energy an genetic modifications due to the lack of understanding.’ This sentence is not clear. Please correct/rephrase it.
- Microarray analysis – ‘….by using and bio-analyzer….’. Remove ‘and’ in the sentence.
- Discussion – ‘IC50 concentrations for both TiB2…….’, this sentence is not clear. Please rephrase it.

Author Response
- This article is about ‘titanium diboride and zinc borate nanoparticles’; however there is not much information on NPs. Introduction contains information mostly about general titanium diboride and zinc borate compounds. Authors should include information on NPs than general compounds.
Information related to titanium diboride and zinc borate were integrated into the “introduction” part as;
“Such compounds are often desired to have non-toxic properties to allow their full potential in a variety of applications and avoid human health risks. The biocompatibility of titanium-titanium boride (Ti-TiBw) composites for use in medical and dental implants investigated with the results indicating that Ti-TiBw composite is blood biocompatible and not cytotoxic to fibroblast cells (Makau et al. 2013).”
“Moreover, in regards to providing modest antimutagenic potential, zinc borate was discovered to have moderate total antioxidant activity. With its antioxidative and antimutagenic qualities, zinc borate has the potential to be used in medicine (UÄŸur et al. 2019).”
“Despite substantial research on these compounds, research into TiB2 and Zn3BO6 compounds for in vitro biological activities was limited and, to the authors’ knowledge, nothing is known about their biological properties and cytotoxic activities which has prevented their usage in bio-applications thus far.”
- Materials and Methods: In the sub-section ‘Characterization of Nanoparticles’, what is the amount of ‘pure ethanol’ and ‘oleic acid’ used?
Information was added to the methods part as;
“Zinc borate (Zn3BO6) nanoparticles were synthesized in a batch reactor by mixing 25 ml of distilled water, 7.5 g of boric acid (H3BO3), 2.846 g of zinc oxide (ZnO), 10 ml of pure ethanol and 3g of oleic acid in a 250 ml three-necked round bottom container.”
- Materials and Methods – Cell cultures and exposure – Please mention the solvent used to dissolve the NPs.
Information was integrated to the part as;
“10 mg of each nanoparticle was placed into polypropylene tubes and dispersed in Milli-Q water (10 ml, at 25 °C) using a sonification chamber. The total sonication time was 40 min (ultrasonication was performed for 20 min). To prevent heating during ultrasonic dispersion the tubes were kept at 0 °C in ice-water bath. After autoclaving of 10 mg/ml NP/Milli-Q dispersions, the stock dispersions were stored in dark at autoclaved and stored as stock dispersions at 4 °C.”
- Materials and Methods – NR uptake assay – Please include the treatment duration of NPs to the cells.
Treatment duration was shown in “Cell cultures and NPs applications“ part as;
“TiB2 and Zn3BO6 NPs were applied to the cell cultures for 72 hours at wide spectrum of concentrations (0.625 mg/L to 1280 mg/L) in triple repeats.”
- Materials and Methods – Microarray analyses – There is no information on the treatment duration and concentration of each NPs used for this analysis.
Information was integrated to the “Microarray analyses” part as:
“LC50 concentrations of NPs (Zn3BO6: 66.50 mg/L and TiB2: 104.34 mg/L) were applied to the cell cultured and total RNA isolation procedure was performed at the end of 72 hours duration.”
- XRD, SEM and TEM images in Figures 1 and 2 must be presented separately (No overlapped images). Images shown in Figures 1B and 2B are not clear. Authors must include high-resolution images for these figures. Further, authors claim ‘highly granular structure’ for the SEM images of ZN3BO6 NPs; however there is no granular structure in the image. Please correct it or replace the image.
Images were separated and magnifications were modified. Sentences were modified and corrected according to recommendations.
- Authors used ‘LC50’ while presenting in vitro cytotoxicity results and the term ‘IC50’ was used in the discussion section. It should be ‘IC50’ throughout the manuscript. Also, authors should include the Figure numbers while presenting the results.
LC50s were modified into IC50 throughout the manuscript and figure numbers were added.
- Authors must clarify which cytotoxicity data was used to calculate LC50/IC50 values for both the NPs. It is better to include IC50 values for each NPs derived from individual cytotoxicity assays.
IC50 calculations were very close to each other so, MTT analysis results were used to calculate IC50 values. Information was added to the manuscript as:
“IC50 values were calculated using probit analysis from the results obtained from the MTT assay”
- ‘According to DAVID annotation….’, reference is missing. Please include suitable reference for the analysis used.
Phrase was corrected as;
“According to STRING (Protein-Protein Interaction Networks Functional Enrichment Analysis, https://string-db.org/) analysis; DNA metabolic pathways were most affected in the cell.”
- Authors should mention Table 1 and 2 while presenting the results.
Table 1 and Table 2 already were mentioned in the results part as;
“The immune system, defense mechanisms and many transcription factor pathways were affected by TiB2 treatment. Considering the number of affected genes, it was observed that there were significant changes in the expressions of genes involved in the immune system (Table 1 and Figure 5). Table 1 lists the 50 genes with the highest expression levels in lung epithelial cells treated with TiB2 NPs.”
“Again, after treatment with Zn3BO6 NPs, cell cycle in cells, cancer pathways and cytokine-receptor molecule regulation are affected. Table 2 presents 50 genes with the highest expression levels in lung epithelial cells treated with Zn3BO6 NPs.”
- What is the rationale for using HPAEpiC cells for this study? Please explain.
Sentences were added to the discussion part as:
“Studies have claimed that inhalation is the most common route for NPs exposure and molecules are transferred from upper respiratory tract to bronchioles by electrostatic force of the air. Thus, the HPAEpiC cell line is one of the most suitable cell culture models to study NPs toxicity that can easily be contacted through the air (Yah et al. 2012).”
- Discussion – ‘As a conclusion, TiB2 and Zn3BO6……’, this statement needs to be modified/corrected as authors performed cytotoxicity studies in only one cell line which do not reflect complete toxicity profile of these NPs.
Sentence was modified according to the reviewer’s recommendations as;
“As a conclusion, TiB2 and Zn3BO6 NPs were investigated as low toxic compounds when exposed to human pulmonary alveolar cell culture that might not rise dramatic side effects under certain conditions. On the other hand, further animal studies should be performed before recommending TiB2 and Zn3BO6 NPs as biocompatible materials.”
Minor:
- ‘…nuclear energy an genetic modifications due to the lack of understanding.’ This sentence is not clear. Please correct/rephrase it.
The sentence was rephrased as;
“On the other hand, society has many concerns about nanomaterials applications due to the lack of understanding in different fields related to nanoparticle uses such as nuclear medicine and genetic modifications”
- Microarray analysis – ‘….by using and bio-analyzer….’. Remove ‘and’ in the sentence.
Sentence was corrected.
- Discussion – ‘IC50 concentrations for both TiB2…….’, this sentence is not clear. Please rephrase it.
Sentence was rephrased as;
“IC50 concentrations of both TiB2 and Zn3BO6 NPs were applied to the cell cultures for 72 hours and at the end of the incubation period microarray analyses were performed by isolating total RNAs from the cell cultures”

Round 2
Reviewer 1 Report
The authors answered my questions and I have no further questions.
Reviewer 2 Report
Manuscript ID: materials-1639277-peer-review-v2
Authors addressed all the concerns with modifications in the text, figures and revised the manuscript accordingly. Authors provided suitable explanations for the reviewer comments in their response letter. Authors also replaced low-res SEM/TEM images with high-res images and presented as individual images in the figures. I recommend this manuscript for publication.